# Trust in science during the COVID-19 pandemic: A typology of internet users in South Africa

**Anne Reif[1]\***, **Lars Guenther[2,3]**, **Monika Taddicken[4]**, **Peter Weingart[2,5]**

**1** University of Hamburg, Hamburg, Germany, **2** Stellenbosch University, Stellenbosch, South Africa, **3** LMU Munich, Munich, Germany, **4** Technische Universität Braunschweig, Braunschweig, Germany, **5** University of Bielefeld, Bielefeld, Germany

\* anne.reif@uni-hamburg.de

## Abstract

The COVID-19 pandemic has underscored the importance of public trust in science, particularly in contexts with diverse cultural backgrounds such as South Africa. Given the increasing reliance on digital science communication and the formation of sub-publics, understanding how different groups of individuals perceive and trust science in this unique setting is crucial for developing effective communication strategies and addressing potential challenges to public trust. This study investigates South Africans' trust in science and contact with science, as well as perceived changes in trust in science due to the COVID-19 pandemic. Through a survey among South African online users and latent profile analysis, we identified four population groups (*n* = 1,541) with varying levels of trust and frequency of contact with science: the *fully trusting (30%)*, *highly trusting (35%)*, *moderately trusting (28%)*, and *rather untrusting (8%)*. Differences in the patterns regarding the five theoretically derived dimensions of trust in science, namely expertise, integrity, benevolence, transparency and dialogue orientation, are subtle: Those who trust the most have particularly high values for benevolence; those who trust less place their highest value on expertise. Variations in self-perceived changes in trust in science resulting from the COVID-19 pandemic suggest differences among groups in how they interpret science communication. This could inform the development of targeted communication strategies. Furthermore, the results point to cultural specifics and indicate that the COVID-19 pandemic has been a key event for public trust in science in South Africa.

## 1. Introduction

 The digitalisation of communication media has transformed the concept of *public* [1] into a constellation of co-existing sub-publics, each potentially exposed to diverse content by different communicators both on- and offline [e.g., 2]. Regarding scientific information, online media – including journalistic outlets and social media platforms – have become the most prominent and widely used sources of information

**Data availability statement:** The data set and materials necessary to reproduce the analyses conducted for the article (RScript, SPSS Syntax, summary of open-ended responses) are available on OSF. Link: https://osf.io/qd3cy/overview DOI: 10.17605/OSF.IO/QD3CY.

**Funding:** This work is based on research supported by the South African Research Chairs Initiative of the Department of Science and Technology, together with the National Research Foundation (NRF) of South Africa (Grant Number 93097). The opinions, findings, conclusions and recommendations expressed in this paper are those of the authors, and the NRF does not accept any liability in this regard. Further, the research presented is part of the project 'The trust relationship between science and digitized publics' (TruSDi), funded by the Deutsche Forschungsgemeinschaft (DFG, German Research Foundation) – 456602133. Grant applicants are Lars Guenther (GU 1674/3-1), Anne Reif (RE 4508/1-3) and Monika Taddicken (TA 712/4-1).

**Competing interests:** The authors have declared that no competing interests exist.

[3]. Given the extensive presence and dissemination of information that ranges from high-quality science journalism to possibly misleading content and conspiracy myths, public trust in science may be fundamentally challenged [4–7].

Studying public trust in science using polls has a long-standing tradition (for the USA, see [8]) and is considered especially crucial during crises such as the COVID-19 pandemic [9–11], when the advice of epidemiological and medical experts to policymakers, as well as the degree of scientific uncertainty due to knowledge gaps, received increased public attention and visibility [12]. At the same time, an overwhelming amount of information was available, including false information and misinformation, which has been described as an 'infodemic' [13]. These unique circumstances likely affected public trust in science [see 14], highlighting the need to examine societal groups with diverging opinion [15]. While some individuals' trust in scientific authorities may have been destabilised [16], it may have also been reinforced and strengthened in others. In fact, representative surveys in the UK, USA, Canada and Germany detected an initial increase in trust in science at the onset of the COVID-19 pandemic [17,18]. However, these surveys do not account for sub-group differences, nor do they explore the various reasons underlying trust in science. Through three experimental studies, Borinca et al. [16] provide evidence that interpretations of science communication messages vary depending on individuals' trust in science and the social norms in culturally diverse groups. They conclude that "trust in science serves as a critical lens through which individuals interpret and respond to scientific information, influencing their willingness to adhere to health guidelines, susceptibility to conspiracy beliefs, and collective actions against authoritative health advice" [16].

This study is situated within this context. In light of the heterogeneous nature of online content and user types [19], as well as potential fragmentation of trust in science in times of crisis, we advocate a group-specific approach that considers individuals' reasons for and the degree of trust in science, thereby capturing diverse patterns of trust in science [15], and examining the relation to exposure to scientific information.

Beyond individual factors, recent literature emphasises the importance of exploring trust in science across different cultural settings. Studies have identified variations in levels of trust in science and its dimensions (e.g., scientists' expertise, integrity, benevolence, transparency, dialogue orientation/openness; [20,21]) as well as differences in the relationship between trust in science and the use of science news on social media across countries [22]. Contrasting with predominantly Western-oriented science communication research [23], this study focuses on the non-Western perspective by investigating trust in science in South Africa (ZA) during the COVID-19 pandemic.

South Africans constitute a particularly interesting case due to its vast cultural diversity, which is also reflected in online media use and individuals' relationship with science – highlighting the need for a group-specific analytical approach. The population is particularly young, with high rates of active social media users [24], and a significant proportion of the population endorses science-related conspiracy myths [25].

While large rural areas with people culturally distant from science [26] coexist with urban centres, religiosity remains highly important [27]. Furthermore, ZA exhibits a *unique fingerprint* regarding perceptions of science and technology (S&T). For example, whereas in the USA [28], beliefs in the promises of science are correlated with fewer reservations about S&T; in ZA, perceptions regarding promises and reservations about S&T are positively associated [29–31]. These particularities make ZA an especially compelling case for examining trust in science.

This study aims to identify distinct *trust groups* within this unique cultural context to shed light on different patterns of trust in science among the ZA public. Additionally, we investigate how these groups differ in their contact with science and whether they have perceived changes in trust in science due to the COVID-19 pandemic.

## 1.1. A differentiated view of trust in science

Diverse social science disciplines have studied trust which is why many definitions and approaches co-exist [e.g., 32]. Commonly, research distinguishes between two different conceptualisations of trust [see 33]. One conceptualisation is *behavioural trust* as willingness to make oneself vulnerable [34], thus focusing on aspects of risk and uncertainty [33]. The second strand of research views trust as "confident positive expectations regarding another's conduct" [35] by focusing on *dimensions of trust* [[15,36]; also referred to as trustworthiness perceptions, see [34,37]]. This study focuses on the second conceptualisation of trust and argues for the necessity to consider different dimensions to help developing specific communication strategies [15,34,36]. Nevertheless, rather than further fragmenting trust research, a meta-definition should combine both main ideas in the sense that trust can be understood as positive expectations towards an object of trust in the face of risks [33]. This means that trust is related to the willingness to be vulnerable to the risk that the object of trust does not meet addressed expectations [primarily based on 37–42]. In this study, we follow the understanding that trust in science is a key variable in the relationship between the public (subject of trust) and science (object of trust). We specifically consider *epistemic trust in science* [43], which refers to the dependence on scientific experts' knowledge and the risk of being misinformed. Public trust in science is the public's assumption that science offers reliable (and relevant) information [44]. To ensure this, scientific expertise, integrity and benevolence are required [45], referred to here as dimensions of trust in science. Based on the idea of *public trust* [37,46], communication research has pointed to the increased importance of media experiences with representatives of science, i.e., their transparency and openness to conversations. This is also reflected by calls of science communication research [15,36,47] and practice [48]. Especially against the backdrop of altered media environments, direct communication is increasingly enabled and insights into processes, contexts and backgrounds are provided that thereby become part of public expectations. Thus, the dimensions of transparency and dialogue orientation appear crucial for an epistemic trust in science and experts [21]. Based on sociological as well as communication research literature, trust in science can be placed on different levels including the macro-level (science as a functional system), the meso-level (scientific organisations) and the micro-level [scientists; see [38–40,42]]. This study focuses on the micro-level.

Rather than treating the public as a single entity [15,49], we highlight the importance to distinguish and examine different groups as suggested in current research on public perceptions of S&T that promotes segmentation approaches [for an overview, 50]. Based on either socio-demographic, psychographic or behavioural variables [51], previous studies have found four to six audiences of science communication in Europe or the USA [52–54]. Trust in science has been central for labelling or describing such population segments [54,55], despite neglecting it as a variable identifying the groups. However, considering group-specific effects regarding public trust in science as a key variable has the potential to inform tailored science communication strategies [16] and offers deeper empirical insights into the question of which theoretical types of trust in science are prevalent within the public – an issue extensively debated within science communication literature. For example, individuals' trust in science may primarily be based on their perception of scientists' expertise (*competence trust* [56,57]) or their benevolence (*goodwill trust* [56] or *trust in motives* [57]). Furthermore, from a different perspective, *blind trust* in science and scientific information [58], meaning complete and uncritical confidence, may not be

ideal from a societal perspective. *Informed* or *cognitive trust* [44,59], as well as a general recognition of science and its function, accompanied by a fundamental trust in the performance of science, may be more desirable. In sum, a differentiated view on public trust in science is needed – which means considering various reasons, i.e., dimensions, and levels of trust by a group-specific approach.

In ZA, little is known regarding public trust in science; current research, however, provides relevant information on general perceptions of science. In contrast to Western countries such as the USA, where people are less likely to be concerned about the negative impacts of science the more optimistic they are [28,60], in ZA, beliefs about the promises of science are positively correlated with reservations about it [29,30]. Hence, South Africans with high levels of trust in science may believe the most in promises while also having strong reservations. Furthermore, religiosity plays a significant role in ZA [27], and many individuals are culturally distant from science [26]. These personal characteristics, which influence general attitudes towards science, may also be negatively associated to people's trust in science. Culturally, however, ZA is characterised by a high score on Hofstede's dimension of power distance, indicating that individuals with lower social authority tend to accept and anticipate an uneven distribution of power. Additionally, ZA is considered a collectivist society, emphasising collective needs over individualistic interests [61]. Based on these cultural dimensions, one might expect a high level of public trust in science within ZA. However, given the unique cultural context and the diverse potential assumptions regarding public trust, different sub-groups may exist that vary in their levels of trust in science, as well as in related factors such as attitudes toward science and the importance attributed to religiosity. Focusing on the South African public's view on science during the COVID-19 pandemic, a time of high risk, we ask the following research question (RQ):

*RQ1: Which groups of ZA online users can be distinguished regarding trust in science during the COVID-19 pandemic and how do they differ regarding socio-demographics and attitudes towards science?*

## 1.2. Diverse types of contact with science

*Access points* form the relationship between science and the public [38]. Science communication and different ways of contact with science have, thus, been predicted to be related to trust in science, which is plausible because science communication not only provides information about science but also different trust cues [62]. At the same time, intermediaries of trust complicate the trust relationship between science and the public, as they themselves are objects of trust [37,62]. In public surveys, three main types of contact are commonly considered: direct or first-hand information, information obtained from intermediaries – namely, social agents such as friends and family – or media outlets, including journalistic sources [15,39]. Usually, the public most frequently encounters science-related information through media reporting or by talking to family and friends [63–65]. While direct contact (e.g., talking to scientists, visiting scientific events) is the least frequent, social media have become important sources [e.g., 66]. Nevertheless, people reported that they trusted direct information more than, for example, journalistic media [49,67,68]. For information about COVID-19 in the UK, health sector sources (e.g., scientists/researchers) were the most trusted. Family and friends were trusted more than journalists/the media, who were among the least trusted information sources [69]. Regarding social media news, scholars found either more sceptical attitudes towards it than other sources [70] or that (especially young) users valued social media as a trustworthy source of information [58].

Regarding the link between the frequency of science-related media use and trust in science, the state of research is inconsistent. While some scholars [71] did not find that the frequency of science-related media use predicted trust in science, others [72] showed that internet use was negatively linked to people's perceptions of scientists and trust in the media. Further, more frequent use of social media has been linked to stronger beliefs in topic-related [73] and general conspiracy narratives [58], which may be connected to distrust in science. In contrast, others found a positive correlation between social media news use and trust in science in a study across 20 countries [22].

Previous segmentation studies considered the frequency of use of different types of science communication [74], often to describe the groups in post-hoc analyses [54]. Usually, the higher a group's frequency of contact with science, the more

positive their attitudes and trust in science [54,74]. For ZA, population segments with a higher frequency of contact tend to be more educated and have stronger beliefs about the benefits of science than groups with less intensive contact with science [75]. The use of social media as well as people's trust in different sources of information, however, have not been considered to date. This is particularly interesting because – partly due to a young population – South Africans are characterised by active social media use [24]. And as a county that is described by cultural values of collectivism and power distance [61], social media use may be a particularly important predictor of public trust in science, as suggested by previous research [22]. Considering all access points and trust in them, we address:

*RQ2: How do groups of trust differ regarding their contact with science and trust in these information sources?*

### 1.3. Perceived changes in trust in science

As the first pandemic of its extent "in the era of wide-spread social media" [14] and as a unique time of globalised risks, COVID-19 may have had a negative impact on public trust in science. According to their secondary analysis of the global 2018 Wellcome Trust survey, Eichengreen et al. [14] found that only people who have encountered epidemics between the age of 18–25 trust scientists less (effects on the micro level). Besides age and the severity of the encountered epidemic, results showed that the loss of trust was greater in low-income countries with weaker medical care and connected with reduced exposure to science at school. In contrast to these results, surveys early in the pandemic showed Germans' increased trust [17,76] and stable trust among the US public [77].

ZA has encountered several epidemics in the recent past. In the case of COVID-19, ZA was among the most affected African countries [78]. In the world-wide comparison, the number of infections was high, but mortality rather low [79]. The ZA government reacted rather quickly to the pandemic by a strict lockdown, which soon has been eased [12]. Considering the 'infodemic' and the assumption that media users may interpret scientific information differently depending on their level of trust in science [16], the COVID-19 pandemic may have led to divergent changes in public trust. Some segments of the population with initially high trust might have perceived an increase in their trust in science, while others with low trust may have experienced further decline or destabilisation. In both scenarios, factors such as the transparency of science communication – and the communication of scientific uncertainties – likely played a significant role.

So far, no longitudinal data for actual changes in trust in science for ZA existed at the time of this study's data collection during the COVID-19 pandemic. However, even asking for self-perceived changes may give first insights. After all, the pandemic may have shaped respondents' answers in this study. Thus, information on perceived changes may help to further characterise and interpret which types of trust in science are prevalent among the groups. Therefore, we want to investigate:

*RQ3: How do groups of trust differ in the perceptions of and reasons for changes in trust in science due to the COVID-19 pandemic?*

## 2. Methods

### 2.1. Data collection and sample

We conducted an online survey among South Africans over 18 years old from November 11 to December 7 of 2020 (the beginning of the second COVID-19 wave). We anticipated unequal group sizes with some groups comprising as little as 5–10% of the total sample. Based on these considerations, we set a target total sample size of $n = 1500$ to ensure sufficient statistical power for detecting small to medium-sized differences across up to seven groups. This sample size also allowed for adequate representation of smaller groups, with an expected minimum of 75–150 participants per group. It facilitates comprehensive descriptive and comparative analyses based on various characteristics.

Participants were recruited via an online access panel by *Ask Afrika*. Members of the panel were randomly invited to voluntarily participate, considering quotas for gender, age, province, population group and geographical setting (Table 1).

**Table 1. Socio-demographic information (frequencies, percentages) for total sample and quota plan.**

| | | Sample | | Quota plan |
|---|---|---|---|---|
| | | *n* | % | % |
| Gender[1] | Female | 913 | 56 | 50 |
| | Male | 711 | 44 | 50 |
| Population group[1] | Black | 1,185 | 73 | 80 |
| | White | 213 | 13 | 8 |
| | Coloured | 162 | 10 | 9 |
| | Indian/Asian | 64 | 4 | 3 |
| Age (*M*=34.17; *SD*=11.24)[1] | 18–24 | 387 | 24 | 23 |
| | 25–34 | 541 | 33 | 26 |
| | 35–44 | 367 | 23 | 20 |
| | 45–54 | 231 | 14 | 13 |
| | 55+ | 98 | 6 | 18 |
| Province[1] | Western Cape | 225 | 14 | 12 |
| | Eastern Cape | 132 | 8 | 11 |
| | Northern Cape | 15 | 1 | 2 |
| | North West | 65 | 4 | 7 |
| | Free State | 80 | 5 | 5 |
| | KwaZulu-Natal | 312 | 19 | 19 |
| | Gauteng | 559 | 34 | 26 |
| | Limpopo | 138 | 9 | 10 |
| | Mpumalanga | 98 | 6 | 8 |
| Geographical setting[1] | Metro | 923 | 57 | 55 |
| | Urban | 429 | 26 | 25 |
| | Rural or tribal | 272 | 17 | 20 |
| Education | Never attended school, attended primary school, or finished with the Grade 9/GET phase | 75 | 5 | |
| | Matric certificate | 525 | 32 | |
| | College certificate | 336 | 21 | |
| | Tertiary (university) education | 688 | 42 | |
| Familiarity with science | Studied science at school | 1,004 | 62 | |
| | Met a scientist personally at least once | 917 | 57 | |
| | Never worked in science | 1,157 | 71 | |
| Household income | Up until R5,000 (~$311) | 279 | 17 | |
| | R5,001–R10,000 (~$311–$622) | 249 | 15 | |
| | R10,001–R20,000 (~$622–$1,245) | 381 | 23 | |
| | R20,001–R30,000 (~$1,245–$1,867) | 255 | 16 | |
| | R30,001–R50,000 (~$1,867–$3,112) | 237 | 15 | |
| | More than R50,000 (~>$3,112) | 141 | 9 | |
| Religiosity (*M*=3.80; *SD*=1.24) | (Rather) not religious | 243 | 15 | |
| | Undecided | 277 | 17 | |
| | (Very) religious | 1,064 | 67 | |

*n*=1,624; [1]variables considered in quota plan.

Informed written consent was obtained by Ask Afrika. Participants had the autonomy to choose whether they wished to complete our survey and were free to end the survey at any time. The survey and our exploratory approach have not been preregistered but received ethical approval from the Research Ethics Committee Social, Behavioural and Education Research of Stellenbosch University (SU project number: 19084).

To achieve the desired sample size, the quotas for the characteristics of hard-to-reach individuals were slightly relaxed. The sample ($n = 1,624$; Table 1) is not representative for the general ZA population. Fewer respondents from the oldest age group were recruited compared to official ZA statistics [80]. Women and white individuals as well as some provinces are overrepresented. The sample comprises many educated people, which may have resulted from using an online survey [81].

## 2.2. Material and measures

The questionnaire was administered in English and employed five-point rating scales, including an additional option: 'I don't know/no response'. The development of the questionnaire involved an intensive, iterative process involving all authors, followed by testing, refinement, and finalisation through multiple feedback rounds with colleagues from South Africa. Validated scales were employed wherever possible. The primary focus was on assessing trust in science and the contact with science based on theoretical ideas [15] and aligned with a comparable project conducted in Germany [21,82].

### 2.2.1. Trust in science and other institutions.
Different theoretical understandings of trust exist which is acknowledged by this study. Based on a theoretical model [15], we distinguished between three levels of trust in science and five dimensions [see also 21]. First, trust in science on the macro-, meso- and micro-level was assessed with direct measures (see Table 2). Respondents were also asked to directly indicate their trust in other institutions and social elites (politics, the media, religion/the church, the military) and in other people in general to give further hints about the types of trust in science prevalent among the groups. We then captured the agreement to the different reasons for respondents' trust in scientists, with two to three items per dimension (expertise, integrity, benevolence, transparency and dialogue orientation, Table 2) as our main focus of the survey and to be used to identify different groups of trust. Our scale is similar to a recently developed scale [21] that considers and combines items and ideas of established instruments (in particular: [36,45,66,85]). We conducted a confirmatory factor analysis (see Tables A and B in S1 Appendix) because the distinction of the five dimensions was based on theoretical ideas [15] and has been empirically tested, albeit only within a German sample [21]. We tested the five-factor solution against several alternative models with fewer factors. First, a single-factor model was evaluated. Subsequently, we examined a two-factor solution suggested by exploratory factor analysis (EFA) with varimax rotation (KMO = .94, $\chi^2 = 9548.48$, $df = 66$, $p < .001$, $h^2 = .40$–76). In this two-factor solution, the first factor explained 55% of the variance and comprised items measuring benevolence, transparency and dialogue orientation. The second factor included all items related to expertise and integrity and explained approximately 8% of the variance. Despite two double-loadings, all expertise and integrity items were retained for comparison between the two-factor and five-factor solutions. Lastly, we tested a four-dimensional model used by previous research [36] against our five-factor model. The five-factor model demonstrated a significantly better fit than all alternative models, revealing acceptable fit indices, with the exception of the RMSEA, which was slightly above the conventional cut-off of .06 (RMSEA = .07[.06,.08]). We also developed two items to measure behavioural trust as willingness to be vulnerable [84] to engage with the current academic discourse [36]: 'I would feel comfortable to rely on scientists' efforts to find solutions to important problems, even if I could not monitor their actions' and 'I would be willing to let scientists have complete control over the future of our society' ($r = .54$). Furthermore, direct measures are applied to different levels to allow a comparison of the micro-, meso- and macro-level. Lastly, in an open-ended question, respondents were asked to elaborate on the extent to which they had perceived a change in their trust in science due to the COVID-19 pandemic (RQ3).

**Table 2. Measures of trust in science and descriptive statistics (mean, standard deviation).**

| | Items & Scale | M (SD) |
|---|---|---|
| Levels of trust (α = .88) | How much do you trust …<br>1 'do not trust at all' to 5 'trust a great deal' | |
| Macro (system) | … science | 3.97 (.97) |
| Meso (organisation) | … scientists at universities and research institutes. | 4.07 (.97) |
| | … scientists in private companies/industry. | 3.86 (1.07) |
| Micro (individuals) | … scientists in general. | 3.93 (1.02) |
| Dimensions of trust | Scientists can be trusted because they …<br>1 'strongly disagree' to 5 'strongly agree' | |
| *Expertise (α = .72)* | … are real experts in their particular fields.[1] | 4.09 (.99) |
| | … rarely make mistakes.[1] | 3.13 (1.27) |
| | … check each other's results before publishing them.[2] | 3.96 (1.08) |
| *Integrity (α = .77)* | … adhere to strict rules and standards in their work.[1] | 3.93 (1.07) |
| | … do not adjust their results for strategic and financial reasons or to please others' expectations.[1] | 3.52 (1.22) |
| *Benevolence (α = .79)* | … work for the common good.[1] | 3.88 (1.07) |
| | … would not purposely bring harm to others.[3] | 3.58 (1.25) |
| *Transparency (α = .83)* | … regularly inform the public about relevant and important results of their research. | 3.64 (1.16) |
| | … do not conceal details of their research. | 3.36 (1.23) |
| | … explain scientific information in a comprehensible way. | 3.71 (1.13) |
| *Dialogue orientation (α = .79)* | … listen to public opinions on their topics and research.[1,4] | 3.41 (1.27) |
| | … do not shy away from public discourse and participation. | 3.63 (1.18) |
| Willingness to be vulnerable (α = .70) | How much do you agree with the following statements?<br>1 'strongly disagree' to 5 'strongly agree' | 3.33 (1.16) |
| | I would feel comfortable to rely on scientists' efforts to find solutions to important problems, even if I could not monitor their actions. | 3.59 (1.22) |
| | I would be willing to let scientists have complete control over the future of our society. | 3.07 (1.40) |

*n* = 1,555−1,610; Variables in italics were used for latent profile analysis. [1]based on [66]; [2] based on [83]; [3]based on [84]; [4]based on [63]. See [21] for the recently developed Publlic Trust in Science Scale.

**2.2.2. Contact with science.** Four types of contact with science were considered (RQ2; [15]). After an open-ended question about preferred sources, we presented a list of sources, asking respondents to report their frequency of use on a scale from 1, 'never' to 5, 'very often' [65,83]. Measures of direct contact included four items (α = .84, e.g., conversations with scientists). In addition, we asked whether respondents have met a scientist in person ('yes'/'no'/ 'I don't know' excluded as missing) and if they are or have been working in science themselves ('Yes, currently'/ 'No, but I used to'/ 'No, never'). Contact with science via social agents was captured using one item about the frequency of conversations with others, such as family, colleagues or friends. Five items measured contact via journalistic media (α = .83, e.g., TV, online journalistic media), and seven measured contact via social media (α = .85, e.g., social networking sites). As a result of exploratory factor analysis (varimax rotation), three items (non-fiction books, fictional content and homepages of scientific institutions) did not fit the two media use factors. In addition to general media use, we captured online engagement with science [19] as search engine use (single item) and online participation (four items, α = .87, liking, commenting, sharing and publishing science-related content online; [see also 86]).

As forms of contact with science are also objects of trust [62], we considered trust in the contact types. On a scale from 1, 'do not trust at all' to 5, 'trust a great deal', we asked how much the respondents trusted in scientific information from *conversations with scientists directly*; *conversations with others, such as family, colleagues or friends*; *journalistic (online) news media* and *social media in general* [83].

**2.2.3. Interest, knowledge and attitudes.** We measured interest and self-assessed knowledge regarding science in general, scientific methods used to generate knowledge and COVID-19 as a scientific topic on a scale from 1, 'not interested at all'/'know nothing' to 5, 'very interested'/'know a great deal' (based on [68]). We used standard items (e.g., [30,65]) to gauge reservations about S&T or views on its promises. We computed reservations ($\alpha = .45$) and promises indices ($\alpha = .67$) despite weak reliability scores [30]. In addition, we used another standard item – 'Whenever science and religion conflict, religion is always right' [87].

## 2.3. Procedure

After the first questions regarding socio-demographics (Table 1), the questionnaire captured interest, attitudes and knowledge, contact with and trust in science or other institutions. The survey ended with questions regarding religiosity, educational attainment, political orientation and monthly household income. Throughout the survey, we made comprehensive efforts to ensure high response quality. Particular emphasis was placed on clear instructions for questionnaire completion. Additionally, open-ended questions were incorporated regularly to enhance engagement and stimulate thoughtful responses. These strategies aimed to maintain participant interest and attentiveness, thereby promoting the overall quality of the data collected.

## 2.4. Analyses

Upon data cleansing, the dataset was screened for implausible response patterns prior to analysis. To answer RQ1, we conducted latent profile analysis (LPA) in *RStudio* using the package *tidyLPA* (model 1) with mean indices of expertise, integrity, benevolence, transparency and dialogue orientation (see Table C in S1 Appendix for correlation table). The aim was to examine which trust groups exist varying in their patterns of dimensions and, thus, reasons to trust in scientists. We chose LPA as an exploratory method of data analysis because we employed continuous data. In order to avoid imputation, 83 cases were excluded due to missing values resulting in a final sample of $n = 1,541$. We identified the appropriate number of groups with the help of fit indices and elbow criterion (see Table D and Figure A in S1 Appendix). We stepwise increased the number of groups to be calculated until the best solution was determined. An analytic hierarchy process, based on the fit indices Akaike information criteria (AIC), Approximate weight of evidence (AWE), Bayesian information criterion (BIC) and Kullback information criterion (KIC), suggested four groups as the best solution. Therefore, we chose this solution in which each subject was assigned to one of four profiles based on posterior group membership probabilities. An additional discriminant analysis in SPSS revealed 96% of correctly predicted cases for those four groups. No group was smaller than 5% of the total sample size ($n = 116–532$).

One-way analyses of variance (ANOVA) with post-hoc tests (Bonferroni) were calculated using SPSS to describe the groups regarding all measured variables and to examine statistically significant differences regarding the mean values across groups. First, we focused on group differences regarding trust in science (RQ1, between and within groups), before examining the links between group memberships and the contact types (RQ2). For RQ3, we coded the open-ended question on perceived changes of trust in science due to the COVID-19 pandemic according to the direction of change addressed (*positive*, *negative*, *mixed/unspecific*, *no/not much change, no reference*), to test for group-specific differences (*chi*-squared test). The intercoder reliability for two independent coders and $n = 150$ was good ($\kappa = .74$). In cases where the two coders' coding did not initially overlap, the cases were discussed until consensus was reached. All remaining cases were coded by a single coder. To give a deeper insight into the material, we structured and systematically and thematically summarised the aspects regarding positive and negative changes, excluding statements without references to a change in trust, e.g., general praise or critique. We extracted prime examples to use as illustration in the results section.

## 3. Results

### 3.1. The ZA groups of trust in science (RQ1)

The results of the latent profile analysis (Fig 1, Table 3) revealed that, according to the mean values, the groups mainly differ regarding the general levels of trust. Therefore, the group labels are primarily based on general trust levels ranging from highest to lowest trust. We found two groups with high levels of trust in science: *(1) fully trusting* and *(2) highly trusting*. Two groups, however, were very different in their levels of trust: *(3) moderately trusting* and *(4) rather untrusting*. Although the labels were chosen to indicate how much the groups trusted, the analyses also revealed differences within the

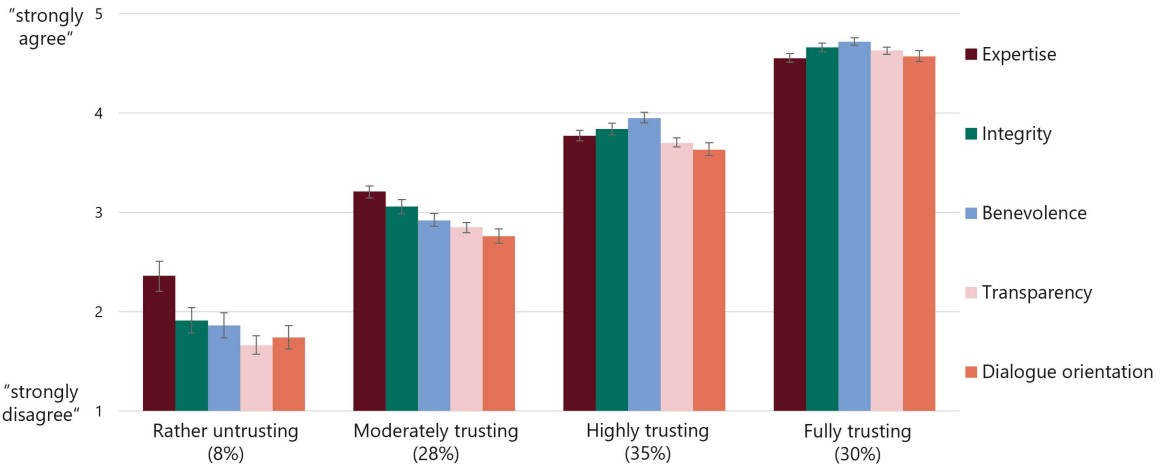

**Fig 1. Patterns for dimensions of trust in scientists of the four trust groups.** *Note. n* = 1,541. Figure depicts mean values and confidence intervals. See Table 3 or Table E in S1 Appendix for exact mean values and standard deviation.

**Table 3. Description of four ZA groups of trust regarding their trust in science (*M* (*SD*)).**

| | 1 Fully trusting (*n* = 465; 30%) | 2 Highly trusting (*n* = 532; 35%) | 3 Moderately trusting (*n* = 428; 28%) | 4 Rather untrusting (*n* = 116; 8%) |
|---|---|---|---|---|
| **Levels of trust[1]** | | | | |
| Macro: science | **4.62 (.60)** | 4.11 (.80) | 3.52 (.89) | 2.91 (1.08) |
| Meso: university | **4.70 (.60)** | 4.17 (.81) | 3.63 (.89) | 2.90 (1.16) |
| Meso: private companies/industry | **4.57 (.60)** | 3.97 (.89) | 3.37 (.94) | 2.43 (1.03) |
| Micro: scientists | **4.64 (.65)** | 4.08 (.80) | 3.40 (.90) | 2.58 (1.06) |
| **Dimensions of trust[2]** | | | | |
| *Expertise* | ***4.55 (.47)a,b*** | *3.77 (.63)a,b* | *3.21 (.63) a,b,c,d* | *2.36 (.83)a,b,c,d* |
| *Integrity* | ***4.66 (.47)a*** | *3.84 (.70)c,d* | *3.06 (.76)a,e,f* | *1.91 (.70)a,e* |
| *Benevolence* | ***4.72 (.43)b,c,d*** | *3.95 (.63)a,e,f* | *2.92 (.68)b,g* | *1.86 (.69)b* |
| *Transparency* | ***4.63 (.40)c*** | *3.70 (.53)c,e* | *2.85 (.53)c,e* | *1.66 (.51)c,e* |
| *Dialogue orientation* | ***4.57 (.61)d*** | *3.63 (.76)b,d,f* | *2.76 (.76)d,f,g* | *1.74 (.63)d* |
| **Willingness to be vulnerable[2]** | **4.20 (.91)** | 3.42 (.95) | 2.70 (.88) | 1.78 (.78) |

*Note. n* = 1,541. *M* = mean, *SD* = standard deviation. Variables in italics were used for latent profile analysis. [1]1 'do not trust at all' to 5 'trust a great deal', [2]1 'strongly disagree' to 5 'strongly agree'. Per variable, all groups differ in the Bonferroni post-hoc test on $p \leq .05$ (between groups). Numbers in bold indicate the highest numbers per variable. Per group, dimensions significantly differing in the Bonferroni post-hoc test on $p \leq .05$ are indicated by matching superscript letters (within groups). See Table E in S1 Appendix for all variables and *F*- and $\chi^2$-values.

groups regarding their patterns for the five trust dimensions measured. The effect sizes indicate that the biggest differences between the groups were regarding transparency ($\eta^2 = .76$), followed by benevolence ($\eta^2 = .68$), integrity ($\eta^2 = .60$), dialogue orientation ($\eta^2 = .59$) and expertise ($\eta^2 = .54$). Additionally, we found a weak positive correlation between respondents' beliefs regarding the promises of science and their reservations against science for the total sample ($r = .14$; $p < .001$).

(1) *The fully trusting (n = 465; 30%)* were South Africans with (almost) complete trust in science across all levels and formed one of the largest groups. They strongly agreed with all dimensions as reasons to trust scientists – most strongly with benevolence and least strongly with expertise and dialogue orientation. Most differences within subjects exist between benevolence and all other dimensions except integrity. They also had the highest willingness to be vulnerable to science and their trust in other institutions was highest among all groups but lower than their trust in science (Table 4). The *fully trusting* had the highest interest in and knowledge about scientific topics and processes. They agreed particularly strongly with the promises of science but were also most critical of it. They were the only group for which a significant positive correlation between beliefs in promises and reservations about science was found ($r = .18$; $p < .001$). Compared to the other groups, the *fully trusting* were slightly older, particularly religious and conservative in their political views. In sum, the *fully trusting* can be described as individuals who have complete trust in scientists' goodwill, but also their scientific performance and communicative abilities. They strongly engage with science both emotionally and intellectually.

(2) *The highly trusting (n = 532; 35%)* had the second-highest level of trust – particularly in academics at universities. They especially valued scientists for their benevolence, followed by their integrity. Within subjects, ratings of benevolence and dialogue orientation differed the most from other dimensions. *Highly trusting* individuals had the second highest willingness to grant science complete control over the societal future. Their trust in religion/the church was high, whereas they were undecided about their trust in the military and media and rather did not trust politics. They were highly interested and knowledgeable. Again, the largest proportion of this group had a university degree. They believed more in science's promises than they had reservations about it; the two contrary beliefs were not significantly correlated ($r = .08$; $p = .061$). In brief, the *highly trusting* are individuals with strong, fundamental trust in scientists and who value scientists' goodwill in particular believing that science brings a better future.

(3) *The moderately trusting (n = 428; 28%)* were the youngest group and had a moderate level of trust in science which was as high as their trust in religion/the church. They trusted scientists from private research institutions and scientists in general the least. Contrary to the first two groups, they tended to agree most with expertise as a reason to trust scientists. Within this group, the ratings of dimensions differed the most; expertise ratings in particular. Their willingness to be vulnerable by giving science complete control is rather low. They agreed slightly more to the promises of science than they had reservations about them; no significant correlation was found ($r = .02$; $p = .695$). Their level of interest in and knowledge about science was moderate. Many of them belonged to the centre of the political spectrum or are rather liberal. All in all, the *moderately trusting* trust in science but not at any costs. They value scientists' expertise and balance science and religion.

(4) *The rather untrusting (n = 116; 8%)* were the minority in the sample and the least trusting with the lowest willingness to give science complete control over the future of society. They were undecided about their trust in the functional system and scientists at universities but tended not to trust scientists from private institutions. They rather disagreed with the trust dimensions as reasons to trust scientists but agreed the most regarding scientists' expertise (difference with all other dimensions). They had a higher trust in religion/the church than in science and (rather) did not trust politics, the media or the military. The *rather untrusting* were the least interested and knowledgeable. They were the only group that agreed more strongly with reservations about science than with its promises. A weak negative correlation was not significant ($r = -.08$; $p = .393$). They were particularly diverse in their political orientation. The *rather untrusting*, in sum, exhibit low overall trust in authorities. If at all, they believe in scientists' expertise and prioritise religion over science.

**Table 4. Continued description of four ZA groups of trust regarding additional perceptions and socio-demographics ($M$ ($SD$)).**

| | 1 Fully trusting ($n$ = 465; 30%) | 2 Highly trusting ($n$ = 532; 35%) | 3 Moderately trusting ($n$ = 428; 28%) | 4 Rather untrusting ($n$ = 116; 8%) |
|---|---|---|---|---|
| **Additional variables** | | | | |
| Trust in politics[1] | **3.03 (1.52)'** | 2.45 (1.28)' | 2.00 (1.06)' | 1.41 (.76)' |
| Trust in the media[1] | **3.66 (1.18)'** | 3.06 (1.09)' | 2.60 (1.02)' | 2.03 (1.03)' |
| Trust in religion/the church[1] | **4.08 (1.21)[a,b,c]** | 3.68 (1.22)[a,d] | 3.48 (1.26)[b,d] | 3.45 (1.50)[c] |
| Trust in the military[1] | **3.77 (1.19)'** | 3.11 (1.16)' | 2.75 (1.11)' | 2.15 (1.17)' |
| Trust in other people in general[1] | **3.50 (1.25)[a,b,c]** | 2.92 (1.05)[a,d,e] | 2.52 (.92)[b,d] | 2.24 (.98)[c,e] |
| Interest in science (Index)[2] | **4.49 (.75)'** | 4.15 (.75)' | 3.81 (.89)' | 3.47 (1.11)' |
| Knowledge (Index)[3] | **4.31 (.79)[a,b,c]** | 3.73 (.78)[a,d,e] | 3.25 (.80)[b,d] | 3.03 (.86)[c,e] |
| Reservations (Index)[4] | **3.61 (1.05)[a,b,c]** | 3.38 (.91)[a,d] | 3.19 (.89)[b,d] | 3.25 (.82)[c] |
| Promises (Index)[4] | **4.48 (.68)'** | 4.03 (.75)' | 3.57 (.85)' | 3.01 (1.06)' |
| Whenever science and religion conflict, religion is always right.[4] | **3.45 (1.55)[a,b]** | 3.17 (1.42)[a] | 2.96 (1.44)[b] | 3.08 (1.57) |
| **Socio-demographics** | | | | |
| Age | **35.26 (10.90)[a]** | 33.34 (11.12)[a] | 33.75 (11.72) | 34.10 (11.28) |
| Gender (female) | 57% | 53% | **58%** | 52% |
| Religiosity | **4.05 (1.18)[a,b,c]** | 3.79 (1.18)[a] | 3.60 (1.27)[b] | 3.64 (1.43)[c] |
| Population group*** | | | | |
| Black | **82%** | 73% | 68% | 70% |
| Coloured | 8% | 10% | **11%** | 8% |
| Indian/Asian | 2% | 3% | **5%** | 2% |
| White | 8% | 13% | 16% | **21%** |
| Geographical location | | | | |
| Metropolis | **24%** | 21% | 20% | 22% |
| City | 34% | **38%** | 36% | 29% |
| Town | 24% | 26% | **28%** | **26%** |
| Rural area | 18% | 15% | 16% | **22%** |
| Level of education | | | | |
| I never attended school | 0% | 0% | 0% | 0% |
| Primary school | 1% | 1% | 0% | **2%** |
| Grade 9/GET phase | **4%** | 3% | 3% | 4% |
| Matriculation certificate | 27% | 34% | **36%** | 29% |
| College | 24% | 19% | 19% | **27%** |
| Tertiary education (University) | **44%** | 43% | 42% | 38% |
| Political orientation*** | | | | |
| Liberal | 26% | 22% | **28%** | 27% |
| Moderate | 25% | 40% | **45%** | 38% |
| Conservative | **49%** | 38% | 27% | 36% |

*Note.* $n$ = 1,541. $M$ = mean, $SD$ = standard deviation. [1] 'do not trust at all' to 5 'trust a great deal', [2] 1 'not interested at all' to 5 'very interested', [3] 1 'know nothing' to 5 'know a great deal', [4] 1 'strongly disagree' to 5 'strongly agree', *Chi-squared* test significant at $p \leq .05$, **$p \leq .01$, ***$p \leq .001$. Groups sharing the same superscripts differ in the Bonferroni post-hoc test on $p \leq .05$ per variable. If all groups significantly differ, it is indicated by '. Numbers in bold indicate the highest numbers per variable. See Table E in S1 Appendix for all variables and $F$- and $\chi^2$-values.

### 3.2. Types of contact with science and trust in information sources (RQ2)

The respondents used diverse sources for science-related information – most frequently online search engines, TV, video platforms, fictional content about science and homepages of scientific institutions. They reported speaking regularly with others; conversations with scientists were least common. The respondents stated to speak even less to scientists than they participate in the online discourse about science (Fig 2 and Table E in S1 Appendix).

For all variables, the groups exhibited the same order in their average frequencies as they previously did in their trust in science. This means, the *fully trusting* used all types of contact most frequently and many of them had even worked (or were working) in science and knew scientists personally. The *highly trusting* reported coming into contact with science the second-most frequently; again, many had met a scientist personally. The *moderately trusting* were less likely and the *rather untrusting* were the least likely to be in contact with science in their everyday lives. The vast majority of them had never worked in science. Thus, the general patterns of information use deviated only slightly per group. This was also reflected in the groups' preferred sources of information. While the *fully trusting* and the *highly trusting* named a variety of preferred sources of information about science, including scientific journal articles and conversations with (acquainted) scientists, the *rather untrusting* mainly stated using 'the internet' and the *moderately trusting* also stated online search engines, websites, TV and books.

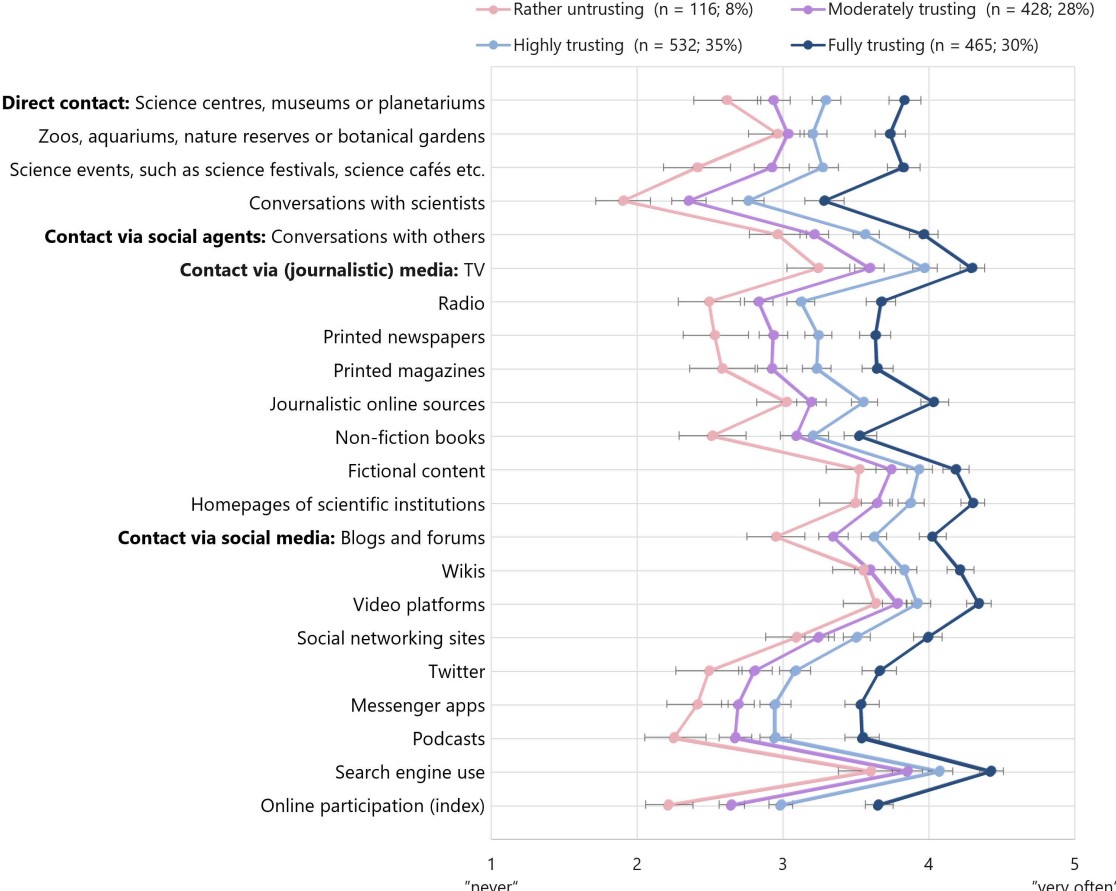

**Fig 2. Information use among the four trust groups.** *Note. n* = 1,541. Figure depicts mean values and confidence intervals. See Table E in S1 Appendix for exact mean values and standard deviation.

Similarly, the groups gradually deviated regarding trust in the different types of contact. Although direct experiences were generally the least frequent, direct information from scientists was trusted the most, followed by information from journalistic media or social contacts. The respondents reported trusting science-related information on social media the least (see Table E in S1 Appendix).

### 3.3. Perceived changes in trust in science (RQ3)

The answers to the mandatory, open-ended question about perceived changes in respondents' trust in science showed that many used the option to share detailed information. For example, over 1,000 respondents provided answers ranging from ten words up to a maximum of 54 words, whereas only 190 participants responded with one to three words. However, less than half of the respondents directly referred to perceived changes in their trust in science in response to the COVID-19 pandemic. Those who did, mainly thought they had experienced a decrease in trust or no/not much change. Differences emerged across the four groups of trust (Fig 3). Most of the *fully trusting* indicated no (significant) change, and despite the second-biggest proportion of them perceiving a loss in trust, it was the lowest of all groups. The percentage of people indicating an increase was the highest. The distribution for the *highly trusting* was similar with only slightly more of them indicating a loss in trust. Compared to the *highly trusting*, a slightly higher proportion of the *moderately trusting* had perceived a negative shift and slightly fewer a positive shift in their trust in science. Proportionally, most of the *rather untrusting* indicated a loss in trust.

Regarding *positive changes* in trust, the first three groups referred to scientists' search for a cure and praised their commitment in the vaccine development.

> My personal trust in scientist[s] has not decreased because of the COVID-19 pandemic; it had in fact heightened. This is because we saw scientist[s] working around the clock day-to-day to ensure that they find a vaccine again[st] the virus to prevent more people getting infected (*highly trusting, 21, female*).

Respondents also stated that science had generated and provided transparent, consistent and accurate information, predictions and recommendations to the public.

> It has actually increased my trust in science. Scientists have been with us throughout this journey during COVID-19. […] They have helped educate our citizens through their findings. Everything that we have learnt about the virus can be credited to scientists (*fully trusting, 28, female*).

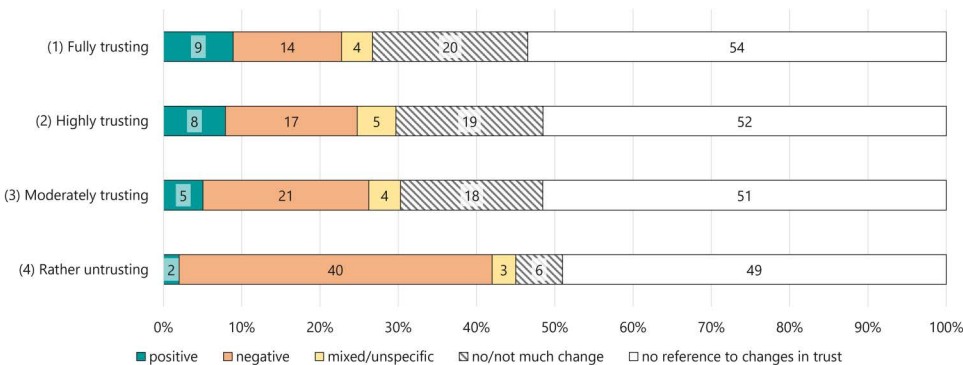

**Fig 3. Changes in trust in science across the four groups (% per group).** *Note.* $\chi^2$ (12, *n* = 1,541) = 57.95, *p* < .001. Classification was based on responses to the open-ended question "*COVID-19 has had an impact on many people in most parts of the world. To what extent has the COVID-19 pandemic affected your personal trust in science? Please elaborate below*".

They also generally praised science and referred to the government as a negative contrast or some stated highly differentiated reasons for a perceived increase in trust and referred to dependence on science for finding solutions but the possibility that scientists could also contradict each other.

The reasons for perceived *negative changes* in trust were more detailed and diverse. Two reasons, however, were prevalent. The first was scientists' perceived lack of competence in finding a cure. The process of vaccine development was criticised for taking too long which was interpreted as scientists' failure.

> The fact that […] so far, there is no confirmed vaccine, or scientific prevention of the pandemic […] has casted a lot of doubt and makes me feel that science is overrated (*highly trusting, 42, male*).

The second one was related to the cause of the virus, which puzzled people across groups but mainly the *moderately trusting*. They addressed rumours about and criticised the lack of clear information about the virus' origins. Most groups claimed the lack of communication and changing, contradictory or false information as reasons for perceiving a loss of trust – first and foremost, the *rather untrusting*.

> Much is not shared with the public, much is not properly explained, and this exacerbates my inherent mistrust of both political and scientific institutions (*rather untrusting, 33, female*).

Other reasons were the perceived lack of science's independence from the government or industry, the effects of the pandemic, missed strategies to foresee and cope with the pandemic or perceived mistakes and lies told by scientists. People of the *rather untrusting* were most critical and even referred to a conspiracy narrative that scientists or other authorities had developed the virus.

> My trust in science has been severely shaken due to the COVID-19 pandemic. I believe that this entire pandemic is a tool that is being used for the governments and other connected organizations around the world to control the population through fear and misinformation. I do not believe that COVID-19 is the deadly virus that we have all been told it is [but] that it is just a smoke-screen for a much larger, more sinister agenda. I believe that scientists created the virus and at the same time created the 'vaccine', and all the talk about the 'vaccine' being in development is a load of garbage (*rather untrusting, 45, female*).

## 4. Discussion

This study investigated public trust in science among internet users in the culturally unique setting of South Africa (ZA) during the COVID-19 pandemic. We identified four groups that differed regarding (the relationship between) trust in science and the frequency of contact with science as well as in the perception of changes in trust due to the pandemic: namely the *fully trusting*, *highly trusting*, *moderately trusting* and *rather untrusting* individuals.

The typology provides empirical insights into the complexity of trust in science as multidimensional variable that varies among different individuals. The *moderately trusting* and the *rather untrusting* highlighted scientists' expertise, which can be interpreted as a form of competence trust [56,57]. In contrast, the *fully trusting* and *highly trusting* based their trust somewhat more on benevolence, which, according to theoretical classifications of trust, may be understood as *goodwill trust* [56,57]. However, it must be acknowledged that we could not verify the existence of *goodwill trust* by individuals with particularly low levels of perceived expertise. A possible explanation for the different patterns of trust in science is that while the *fully trusting* and *highly trusting* may not contest scientists' expertise but emphasise scientists' warmth, the *moderately trusting* and *rather untrusting* may hold on to the stereotype of cold but competent scientists [88] – possibly because of their rare direct experiences with science. The dimensions of transparency and dialogue orientation were also

rated important reasons to trusting scientists by the *fully trusting* and *highly trusting*; less so by the *moderately trusting* and *rather untrusting* individuals. Transparency seems particularly relevant, as it was the dimension by which the groups deviated the most. These standardised results, however, may rather reflect the groups' perception of (a possible lack of) transparent and dialogue-oriented science communication during the pandemic, as suggested by the open-ended answers. While the *fully trusting* and *highly trusting* thought that scientists communicated particularly transparently, the *moderately trusting* and *rather untrusting* criticised the information given as misleading and insufficient. This finding may provide additional support that individuals' trust in science influences how they interpreted means of science communication [16] (or the other way around). Thus, the open-ended questions helped to add context to the standardised measures and highlight the importance of aspects of science communication and scientists' communicative abilities when it comes to trust in science [20, see also 21].

Regarding the levels of trust, we found hardly any differentiations. The respondents only reported a slightly higher trust in scientists working at university compared to those working in the private sector. The groups of *rather untrusting* individual were found to distinguish the most between the levels; the *fully trusting* and *highly trusting* the least which may be a hint to their general high level of trust in authorities that is also reflected in their high trust in the military and politics. Higher values on the direct measures of trust in science on the three different levels were also linked to higher values on the dimensions of trust and a higher willingness to be vulnerable indicating that, although addressing a different angle, they may be similarly sufficient indicators for public trust in science. Unlike the results by previous research [36], we found similar correlations for all dimensions of trust that were weaker in comparison but still strong. On a methodological level of the comparison of measures, we can conclude that all measures of trust are strongly linked, but do not capture the same. Perceptions and behavioural trust are slightly closer related to each other than to the direct measures of trust.

The present study also expands research into the role of different types of contact with science for public trust in science, social media use in particular. The findings revealed a wide range of different information sources used, including high quality journalism on the one hand and social media platforms with possibly misleading content and conspiracy myths on the other hand. However, explicit mentions of conspiracy myths were not as prevalent as expected [25], only referred to by the *rather untrusting*. We confirmed a general tendency for people who trusted science more to come into contact with it more often [see 54,74,75] and to report being more interested and knowledgeable. This contradicts the assumption that a particularly high level of trust can be defined as blind trust. The fully trusting, for whom the probability of blind trust is supposedly high due to their (almost) complete trust, are at the same time also particularly critical regarding the reservations of science rather hinting towards *cognitive trust* [59] or *informed trust* [44]. It has to be noted, that at least for some types of contact with science, a COVID-19 effect may be behind possibly higher frequencies of using scientific information. This may include the importance of online search engines that is highlighted by the results but also the frequent use of websites by scientific organisations.

The *rather untrusting* reported the least frequent contact with science and most often that their trust in (scientific) authorities has been negatively affected by the pandemic – and thus, may indeed have been destabilised and potentially led to further fragmentation as suggested by Borinca et al. [16]. This also suggests that even single past experiences with science and specific content might be more important than measuring frequencies in a standardised way. The *rather untrusting* stated that insufficient information provided was an important reason for their loss of trust in science. As they referred to conspiracy narratives, it is plausible that they came across such content online. Nevertheless, other groups also perceived a decrease in trust in science, despite high actual trust levels. Our empirical findings, therefore, provide evidence that the COVID-19 pandemic has been a key event for public trust in science [see 9,14,17,89]. However, these results may also illustrate that perceived changes that were asked retrospectively are not necessarily accurate to actual changes, or at least that answers to open-ended questions do not have to correspond to standardised measures. Thus, in contrast to the possibility that *rather untrusting* are primarily individuals whose trust has been negatively affected by the pandemic, the results might also indicate that the pandemic consolidated their already low trust in science. The frequent

use of providing detailed information to the open-ended question indicates strong communication needs regarding the pandemic.

We examined public trust in science within the specific cultural context of ZA, discovering a high level of trust and no significant group of individuals untrusting of science. The results, however, may vary in different cultural settings [for a comparison with Germany, see [82,90]. For example, the patterns for the dimensions of trust in science on the micro-level discovered here for the four groups may differ based on cultural factors. The same may be true for the positive correlations between perceptions of science-related promises and reservations, adding new insights to the 'unique fingerprint' of the ZA public [30]. In contrast to other countries, such as the USA and China, where positive and negative attitudes could be found within one group [91], in ZA, we found a small positive correlation between the attitudinal variables for the total sample. The typology, however, showed that this positive correlation was only significant for the *fully trusting* individuals, who most strongly believed in promises about S&T by holding the most reservations about it at the same time. Only the *rather untrusting* tended not to trust science and had greater reservations about S&T than optimistic beliefs. Furthermore, the positive connection between trust in science and religiosity is in stark contrast to the USA, where people of Christian faith are significantly less trusting of science than non-religious individuals [8,91]. This means that in ZA, religion and trust in science do not (necessarily) contradict each other [see [27]]. The *fully trusting* were the most religious and politically conservative. Only the *rather untrusting* trusted the church more than science. Also, the strong positive correlation between the two items of the behavioural trust measures (see Table 2) and thus moderately high agreement to the second statement 'I would be willing to let scientists have complete control over the future of our society' may be a cultural characteristic of the ZA public. According to Hofstede's cultural dimensions [61], the ZA public is described by high values of collectivism and power distance and, thus, more likely to perceive that scientists are working for the common good and less likely to question authority [90]. It may also be an effect caused by the time of data collection early on during the COVID-19 pandemic before the vaccine-rollout in a country that had a comparatively low mortality rate [12]. However, this particular item of the behavioural trust measures may also be problematic, as it implies that scientists should be responsible for political decisions. Particularly the *rather untrusting* have criticised the politicisation of science during the pandemic which may explain the group's low agreement to this item. Future research should advance the measures, e.g., by focusing on laypeople trusting and considering scientists' recommendations in their individual decision-making processes. Furthermore, the high significance of social media use found may be related to the young population in ZA or a reproduction of results found in previous research indicating a particularly important role of social media use as a predictor of public trust in science across collectivist countries high in power distance [22]. However, it could also be a sampling effect. Going beyond different contact forms for science, such as online platforms, only limited assertions about the specific contact, key experiences and media content used can be given.

## 5. Limitations

Some additional limitations will be discussed in the following. The fact that the sample is highly educated with many white individuals may have contributed to a bias towards more positive beliefs and stronger trust in science. One reason for the sample structure is that the research team could not provide the questionnaire in all official languages of ZA, but was limited to the English version. The result may also be influenced by self-selection bias and is inherently linked to the method of surveying individuals via online access panels. It is reasonable to assume that individuals who voluntarily register for such panels are more likely to hold positive attitudes towards science.

The operationalisation of trust in science represents an attempt to improve measurements. However, questions on the meso-level can be critically discussed to be overlapping with the micro-level, despite alignment with the recognized public surveys [see 15,68,92] for the compatibility of our measures with established procedures. On the micro-level, it could be argued that the wording of the item 'trust in scientists in general' aligns more closely with the macro-level, as it can be interpreted as a form of collective rather than interpersonal trust. Specifically, it pertains to trust in scientists as

representatives of the institution of science rather than trust in individual scientists. Moreover, our decision to apply the five dimensions exclusively at what we understand as micro-level warrants further discussion. Applications to the meso- and macro-levels are indeed plausible. Nevertheless, we contend that the micro-level is particularly crucial, as, theoretically, it is on the micro-level that institutional trust can be both developed and altered [38]. Although our data hints towards five dimensions of trust in science, we also have to acknowledge that different theoretical assumptions about the number of dimensions exist that could have been applied [26]. Furthermore, capturing the rather vague 'direct' measures has likely captured some of the variance of the five dimensions as well as some of the variance of the behavioural trust measures.

The use of Latent Profile Analysis (LPA) could also be subject to critical scrutiny. We have addressed this limitation by comparing the results with additional clustering methods. Another concern is the selection of variables used for forming groups. Since the focus is on trust groups, we chose to cluster the groups based on the five dimensions of trust in science and describe the groups' patterns of trust. Different approaches might be considered depending on the research focus. The variations in the patterns across dimensions found were unexpectedly subtle as the tendency of all answers to the dimensions reflected the general descending order of trust groups from *highly trusting* to *rather untrusting*. Unlike expected, we did not identify groups of individuals highly trusting in expertise but moderately trusting in transparency, for instance. This is why somewhat simplistic labels were chosen for the four groups. Nevertheless, interesting patterns were found regarding the five dimensions that indicate the usefulness of the LPA. The result that the *moderately trusting* and the *rather untrusting* groups emphasised scientists' expertise the most, while the *fully trusting* and *highly trusting* groups especially highlighted scientists' benevolence in their evaluations. This is a key finding of this study. Certainly, however, response biases or bad compliance among participants (straight clicking) cannot be completely ruled out, although we made comprehensive efforts to ensure high response quality with measures throughout the survey including incorporating open-ended questions and careful instructions how to respond to each survey question. For example, regarding the open-ended question about perceived changes in trust in science due to the COVID-19 pandemic, the majority of respondents provided detailed answers and on the final page of the survey, over 500 respondents even took the time to add comments to the survey or on the topic This demonstrates a high level of engagement and attentiveness until the end of the survey. Nevertheless, future studies should consider including additional attention checks to further ensure and enhance data quality.

Although the practical relevance of measuring five dimensions instead of a superficial single item measure can be questioned in light of the subtle differences, we believe that our approach provides an interesting added value. And unlike regression analyses, which typically examine predictors of trust in science [e.g., 22,71], our methodological approach acknowledges the assumption that more trust in science is not necessarily better. Instead, it adopts a more value-neutral assumption that trust varies individually and the reasons for this variation are manifold. However, comparing the groups for possible differences in regression analyses, could be a beneficial next step of research (e.g., concerning the relationship between trust and different types of contact with science; [93]).

Lastly, as the design of this study is only cross-sectional, we cannot report on how trust has actually shifted compared to before the pandemic. Asking respondents to self-report how their trust in science has changed using an open-ended question has clear limitations, but also revealed interesting explanations to the groups.

## 6. Conclusion

This study examined perceptions of science in a culturally unique non-Western country and highlights the importance of typologies and longitudinal research for future studies about how and why trust in science changes across different groups of people. Instead of simply identifying and describing different groups of trust in science, we showed how group membership is connected to diverse types of contact with science. These findings have practical implications and can be used to derive effective communication strategies in reaching these groups to potentially stabilise public trust in science. Thus, individual differences in trust in science should be considered in the future when means of science communication are

designed [16]. In the case of the *rather untrusting*, media representations pointing to the expertise of scientists could be perceived as trust-building. Conversely, it may also be promising to focus on interventions that highlight efforts of transparent and dialogue-oriented science communication [20] when the *moderately trusting* and *rather untrusting* are the target groups. Actual group-specific effects, however, need to be investigated experimentally or in longitudinal studies. It also seems profitable to shed more light on the *rather untrusting* using qualitative methods to analyse key experiences and trust cues provided in the received media content.

## Supporting information

**S1 Appendix. Supplemental material to manuscript Trust in science during the COVID-19 pandemic: A typology of internet users in South Africa.**
(DOCX)

## Acknowledgments

We wish to thank Johann Mouton, Marthie van Niekerk, Milandré van Lill, Marina Joubert, Francois van Schalkwyk, and Corlia Meyer from the Centre for Research on Evaluation, Science and Technology (CREST) at Stellenbosch University for commenting on drafts of the survey, and for helping to organise this project. We also wish to thank Justin T. Schröder for commenting on the initial manuscript.

*Author bibliographies*

Anne Reif is a senior researcher at the University of Hamburg, Germany. Her main research interests are in the fields of science communication and digital communication. She focuses on public trust in science as well as public perceptions of climate change in connection to the use of online media.

Lars Guenther is professor of communication science at LMU Munich's Department of Media and Communication in Germany, and research fellow at the Center for Research on Evaluation, Science and Technology (CREST) at Stellenbosch University in South Africa. He is interested into public perceptions of (controversial) science, science and health journalism, trust in science, as well as the public communication about risks and scientific (un)certainty.

Monika Taddicken is professor and heads the Institute for Communication Science at Technische Universität Braunschweig, Germany, a member of the TU9-Alliance. Her main interest is the intersection of digital and science communication. Her research focuses primarily on the user perspective. In addition, she has a strong methodological interest and applies a variety of different empirical methods.

Peter Weingart is Professor emeritus of Sociology, Sociology of Science and Science Policy, Bielefeld University. He is member of the German Academy of Technical Sciences (acatech) and of the Berlin-Brandenburg Academy of Sciences and Humanities (BBAW). 2015–2020 he held the South African Research Chair in Science Communication hosted by CREST, Stellenbosch University. His current research interests are science advice to politics, science – media interrelation, the role of science communication in democratic publics.

## Author contributions

**Conceptualization:** Anne Reif, Lars Guenther, Monika Taddicken, Peter Weingart.

**Data curation:** Anne Reif, Lars Guenther, Monika Taddicken, Peter Weingart.

**Formal analysis:** Anne Reif, Lars Guenther.

**Funding acquisition:** Anne Reif, Lars Guenther, Monika Taddicken, Peter Weingart.

**Methodology:** Anne Reif, Lars Guenther, Monika Taddicken, Peter Weingart.

**Project administration:** Anne Reif, Lars Guenther.

**Supervision:** Monika Taddicken, Peter Weingart.

**Validation:** Anne Reif, Lars Guenther.

**Visualization:** Anne Reif.

**Writing – original draft:** Anne Reif.

**Writing – review & editing:** Anne Reif, Lars Guenther, Monika Taddicken, Peter Weingart.

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
