## [Decision Letter · Decision Letter 0]

29 Dec 2024

Dear Dr. Reif,

We look forward to receiving your revised manuscript.

Kind regards,

Felix G. Rebitschek

Academic Editor

PLOS ONE

Journal Requirements:

“This work is based on research supported by the South African Research Chairs Initiative of the Department of Science and Technology, together with the National Research Foundation (NRF) of South Africa (Grant Number 93097). The opinions, findings, conclusions and recommendations expressed in this paper are those of the authors, and the NRF does not accept any liability in this regard. Further, the research presented is part of the project ‘The trust relationship between science and digitized publics’ (TruSDi), funded by the Deutsche Forschungsgemeinschaft (DFG, German Research Foundation) – 456602133. Grant applicants are Lars Guenther (GU 1674/3-1) and Monika Taddicken (TA 712/4-1).”

Additional Editor Comments:

Dear corresponding author,

You have received three expert reports, many of which express a considerable need for elaboration of the present manuscript and at the same time provide you with many helpful suggestions.

Please take them seriously for your revision.

Reviewers' comments:

Reviewer's Responses to Questions

**Comments to the Author**

1. Is the manuscript technically sound, and do the data support the conclusions?

Reviewer #1: Partly

Reviewer #2: Yes

Reviewer #3: Yes

2. Has the statistical analysis been performed appropriately and rigorously?

Reviewer #1: Yes

Reviewer #2: Yes

Reviewer #3: No

3. Have the authors made all data underlying the findings in their manuscript fully available?

Reviewer #1: Yes

Reviewer #2: No

Reviewer #3: No

4. Is the manuscript presented in an intelligible fashion and written in standard English?

Reviewer #1: Yes

Reviewer #2: Yes

Reviewer #3: No

Reviewer #1: Dear authors,

thank you very much for providing such an interesting manuscript, that I really appreciated to review. The manuscript focuses on the identification of subgroups within the South African population with diverging levels of trust in science measured by different reasons to trust science. Based on recent literature, the authors derive a five factors solution for measuring trust in science. They identify four distinct subgroups among their sample, that mainly differed in their levels on trust in science but had comparable patterns of reasons to trust and on other variables (e.g. contact with science through different mediators).

In its present form, I would not recommend the manuscript for acceptance. I propose that major revisions be undertaken.

The current version of the manuscript attempts to incorporate too many different thematic aspects, resulting in the loss of a clear focus early on. It remains unclear to the reader what the main objective of this paper is: differentiating between online and offline users, comparing the ZA sample with findings from WEIRD states, developing a new instrument for measuring trust in science at the micro level, distinguishing between trust in science at the micro-, meso-, and macro-levels, classifying those who trust, empirically testing the theoretical model by Reif & Guenther (2022) (see Supplemental Figure 1), including mediation analyses of mediated contact with trust in science, or tracking changes in trust over the course of COVID-19?

The manuscript requires greater coherence and clarity, with a more precise focus on the most important research questions. While these points are individually interesting, attempting to empirically address all of them at once makes the analysis less comprehensible.

I would like to highlight the following points that particularly caught my attention:

Content

The authors describe their sample as “online users” and “digitized public”. Yet, it is not clear why the authors refer to this notion and how “online users” differ from “non-online users”. The authors should provide more information for the necessity of this distinction and cite literature that suggests fundamental differences between these subpopulations. While it is understandable that, due to the nature of the survey being conducted online, the participants should be online users, this alone does not seem sufficient to justify narrowing the perspective to such an extent.

Given that the scales are primarily designed to measure trust at the micro level, this should be appropriately reflected in the manuscript, with an emphasis on trust in scientists. Particularly in the theoretical background, the current wording gives the impression that trust is being measured at the macro level, where different reasons for trusting science are assessed (e.g. RQ1).

Methodology

a) For measuring trust in science (micro level), the authors employ newly developed scales, which are partly adapted from existing instruments and partly created anew. They indicate that the dimensions include expertise, integrity, benevolence, transparency, and dialogue. Given that these measures are pivotal for the classification of participants in the LPA, the authors should elaborate further on the measurement instrument. Especially, the issue of construct validity has not been sufficiently addressed and is not evident to the reader.

b) The authors also assess the factorial structure of the instrument through confirmatory factor analysis. Given that the discussion highlighted unresolved issues with item allocation, the authors might want to consider employing exploratory factor analysis for further evaluation. For example, the peer review item is classified under expertise, although it could be more fittingly associated with integrity, dialogue, or transparency as aspects of good scientific standards.

Finally, the authors should justify their decision to forgo the use of well-established scales (e.g., Hendriks et al., 2016; Ziegler, Ricarda, & Kremer, Bastian (2023). Wissenschaftsbarometer 2022 - Repräsentative Bevölkerungsumfrage zu Wissenschaft und Forschung in Deutschland. GESIS, Köln. ZA7641 Datenfile Version 1.0.0, https://doi.org/10.4232/1.14077) for assessing trust in scientists.

c) The behavioural measures need to be critically revised. The item 'I would be willing to let scientists have complete control over the future of our society' appears particularly problematic, as it implies that scientists should be responsible for political decisions, unnecessarily politicising the issue. A better approach to capturing the behavioural aspect might be to focus on laypeople trusting and considering scientists' recommendations in their individual decision-making processes.

d) The authors utilise Latent Profile Analysis (LPA) as the classification method, which was conducted based on the levels of trust in science. There are a few points that the authors should briefly address in this regard: LPA estimates the probabilities for each individual to belong to a specific class, highlighting the probabilistic nature of this approach. However, in the manuscript, class memberships are treated as deterministic, thus overlooking the uncertainty in classification. The authors should explain the methodology they employed to arrive at the final solution, where each individual is assigned to exactly one class.

Results

The results imply that the groups differ only in the degree of trust they place in scientists for various reasons. This is also evident in the frequency and manner of their interaction with scientific information and their trust therein. When examining the findings altogether, they reveal intriguing yet sometimes contradictory patterns: for instance, class 1 demonstrates high levels of trust not only in scientists but also in politicians and the military. How can these findings be interpreted?

Furthermore, the overall results prompt two additional questions:

a) Are these findings truly reflective of real differences, or could they be influenced by response biases among participants? It would be advantageous if the authors addressed this more comprehensively and included further indicators of high data quality (e.g., excluding participants with low item variance across multiple variables).

b) Since the results appear to indicate only subtle differences, how practical is the precise measurement of trust in science, scientists, and contact with scientific information for future research from a perspective of parsimony? Do these findings not strongly suggest that a single item could typically provide a sufficient, though possibly superficial, insight into the level of trust?

The authors describe in their results section, that they performed ANOVAs to compare group means across different variables (between-groups). However, when providing more details about the different classes, the authors also refer to mean comparisons within the classes but across these variables (within-groups). It is not feasible for the reader to assess, whether these differences are of descriptive or inference statistical nature (e.g. Section 6.2).

Discussion

The authors are advised to significantly revise the discussion. It is difficult to discern a consistent line of reasoning, as the discussion often veers off, giving the impression that the authors are trying to include too much content in too little space. Moreover, the discussion includes aspects that are more appropriately discussed in other parts of the manuscript: a large portion of the commentary on the groups should be placed in the results section, and the correlations should ideally be introduced in the results section before being elaborated on in the discussion.

Open Science

In light of open science practices, the authors are requested to briefly state whether there was a citable preregistration for the project detailing the hypotheses and analysis steps. Additionally, for the publication, it would be useful to provide a link to the repository that will be made accessible after the project's completion, which will allow readers to access the data, materials, and scripts.

Language

The authors are requested to carefully review the manuscript once more. Additionally, they should consider whether some of the statements are made with the appropriate level of rigor and whether the existing evidence supports them, e.g.:

- L29ff. “These surveys, however, mainly applied direct single-item measures of public trust in science that neglect the assumption that depending on cultural and individual beliefs and experiences, trust in science may be based on different reasons.”

- L53/L54 sentence structure

- L118-120 sentence structure

- L544ff. “Unlike regression analyses, which typically examine predictors of trust in science, our methodological approach acknowledges the assumption that more trust in science is not necessarily better. Instead, it adopts a more value-neutral assumption that trust varies individually and the reasons for this variation are manifold.”

Others

- L161 – 4 / see also L233/234 (Source links)

- L220 – RMSEA above .60?

- Table 3., Table 4., Supplemental Table 5. – Last Column – “5”? Rather untrusting - should be "4"

In conclusion, I would like to thank the authors once again for this intriguing manuscript and hope that the points raised in this review are clear, comprehensible, and useful. I look forward to reading the final version of the manuscript.

Reviewer #2: The paper, "Trust in Science During the COVID-19 Pandemic: A Typology of Internet Users in South Africa," explores trust in science among South African internet users during the COVID-19 pandemic. Using latent profile analysis, it identifies four groups with varying trust levels—fully trusting, highly trusting, moderately trusting, and rather untrusting. The paper examines trust dimensions (e.g., expertise, benevolence, transparency) and their relationships with contact types, such as direct engagement with scientists or social media. Trust patterns are shown to be influenced by socio-demographics, cultural factors, and personal experiences during the pandemic. The authors highlight the need for group-specific communication strategies to foster trust, particularly among groups with lower trust levels.

While the paper is relevant and engaging, the following revisions are necessary for it to be considered for potential publication in PLOS ONE:

1. The introduction needs further development to provide clearer context and better integration of existing literature.

2. The results section requires clarification and more detailed explanations, particularly regarding latent profile analysis and trust dimensions.

3. The discussion section would benefit from enhanced theoretical integration, contextual specificity, and articulation of broader implications.

Below are specific comments for further refinement:

1. Abstract: The concept of "infodemic" may not be easily understood by a lay audience. Consider providing a brief definition (e.g., an overabundance of information and misinformation related to COVID-19).

2. Page 2: The discussion of group-specific approaches to understanding trust in science emphasizes the role of cultural and individual beliefs in shaping trust patterns. However, it overlooks relevant work by Borinca et al. (2024), which examines trust in science among individuals with varying trust levels (low vs. high) and incorporates both WEIRD (Irish) and non-WEIRD (Kosovo Albanian) samples. This paper experimentally manipulates social norms as a communication strategy and demonstrates how compliance with health advice influences trust, solidarity, and behavior during the COVID-19 pandemic.

o Including this reference would enhance the discussion by providing evidence on how tailored communication strategies and social norms impact trust in culturally diverse groups.

o Beyond the fact that your paper is cross-sectional while Borinca et al.’s is experimental, clarifying how your work uniquely contributes to this area—perhaps through a focus on specific cultural contexts or unique trust mechanisms—would strengthen the discussion.

Reference:

Borinca, I., Griffin, S. M., McMahon, G., Maher, P., & Muldoon, O. T. (2024). Nudging (dis)trust in science: Exploring the interplay of social norms and scientific trust during public health crises. Journal of Applied Social Psychology, 54(8), 487–504.

3. South African Context: The discussion of trust and contact during COVID-19 lacks sufficient context specific to South Africa. Adding examples or background on trust dynamics in South Africa would provide a more comprehensive understanding, particularly as this is a non-WEIRD setting.

4. Theoretical Framework: The distinction between macro, meso, and micro levels of trust is a valuable framework. However, the categorization of "trust in scientists in general" as micro-level trust seems unclear. Typically, micro-level trust involves interpersonal trust in specific individuals, whereas trust in "scientists in general" appears more collective and might align better with meso or macro levels. Clarifying this would enhance the theoretical framework’s precision and interpretability.

5. Trust Profiles:

o The labels for the four groups (e.g., fully trusting, highly trusting) are somewhat simplistic. Could an individual be highly trusting in expertise but moderately trusting in transparency? Acknowledging potential overlaps or ambiguities would add depth to the findings.

o Specify how these profiles were quantitatively defined. Were the cutoffs for the trust levels determined based on a statistical criterion or subjectively labeled?

6. Trust Dimensions: While the authors note differences in trust dimensions across groups, more elaboration is needed. For example, which dimensions (e.g., expertise, transparency, benevolence) most strongly influenced group distinctions?

7. Contact Types:

o The results indicate fewer individuals in the "moderately trusting" and "rather untrusting" groups had personal contact with scientists. Was personal contact measured as binary (e.g., yes/no) or continuous (e.g., frequency of interactions)?

o Were specific types of contact (e.g., formal talks, informal discussions) associated with higher levels of trust? Clarifying this would provide actionable insights for science communication strategies.

8. Behavioral Trust Measures: The paper includes behavioral trust measures, such as willingness to rely on scientists' efforts or allow them control over societal decisions. Were significant differences observed in these measures across the four groups? Highlighting these differences would reveal the behavioral implications of trust levels. If no differences were found, discussing why these measures did not vary would also be valuable.

9. Open-Ended Responses: The responses to RQ3 on changes in trust during COVID-19 are intriguing but lack clarity on how they were analyzed. Were these responses analyzed systematically (e.g., thematic analysis), or are they illustrative examples? Clarifying this would strengthen the methodological rigor.

10. General Discussion:

o The general discussion effectively summarizes the findings but would benefit from explicitly linking the trust profiles to the macro, meso, and micro levels of trust in the theoretical framework.

o Additionally, the discussion should address how the paper’s findings contribute to or challenge existing theories of trust in science.

11. Broader Implications: While the paper addresses the immediate implications for science communication, consider expanding on broader societal and policy implications. For example, how might these findings inform efforts to build trust in science in contexts with historically low trust, such as post-colonial or non-WEIRD societies? Suggestions for future research, such as longitudinal studies to track changes in trust over time or interventions targeting specific trust dimensions (e.g., transparency, benevolence), would also provide valuable direction.

I wish the best to the authors in revising this important work.

Reviewer #3: This research examines trust in science during COVID-19 in South Africa, and provides typologies of individuals based on their trust levels. The use of a more comprehensive measure of trust in science is commendable.

- Paper needs to be reread thoroughly to improve the syntax in places, and typos (e.g., line 23, should read “focused on this interest”

Methods

- Given this is a scale the authors created, it is unclear why CFA was chosen to test the dimensions of the scale. Given it is a new scale, why wasn’t EFA used?

Results

- Unsure where this conclusion stems from “In sum, the fully trusting can be described as people who think that scientists have it all and who embrace science with heart and mind”.

Discussion

- It would be worth discussing the generalisability of the findings in the Discussion. For instance, this sample seems very highly educated.

**Do you want your identity to be public for this peer review?** For information about this choice, including consent withdrawal, please see our Privacy Policy

Reviewer #1: No

Reviewer #2: No

Reviewer #3: No

---

## [Author Response · Author response to Decision Letter 1]

22 Jul 2025

Please see document: "Response to Reviewers"

---

## [Decision Letter · Decision Letter 1]

30 Sep 2025

Dear Dr. Reif,

Thank you for submitting your manuscript to PLOS ONE. After careful consideration, we feel that it has merit but does not fully meet PLOS ONE’s publication criteria as it currently stands. Therefore, we invite you to submit a revised version of the manuscript that addresses the points raised during the review process.

We look forward to receiving your revised manuscript.

Kind regards,

Felix G. Rebitschek

Academic Editor

PLOS ONE

Journal Requirements:

**Additional Editor Comments:**

Dear Authors,

Thank you very much for your extensive revision that has been well appreciated by the reviewers.

Only a few minors remain.

Abstract

- Please start with a sentence about what motivated the study

- Method: where do "the five dimensions of trust in science" come from?

- "provide clues"  inform?

Introduction

- l.33 "became relevant issues" - please clarify

- l.36-37 reference for the fragmentation

- What is meant by reservations - this needs more explanation, because it is presented in contrast to beliefs - which is a perception about somethings' current state (e.g, vaccine x is effective), beliefs are not attitudes (e.g., disliking vaccines)

- l.108 this study focuses on...

- the paragraphs under "3 diverse types of contact with science" - please clarify the argument linking frequency of use and trust cues of certain media. At the moment pp.7-8 reads a bit back and forth

- l. 219 what is the nature of trust

- RQ3 is not very concrete, what is meant by "HOW" do they differ?

Methods

- The power analysis is missing

- the sample is not representative for what?

- where is the "quota plan"?

- please provide the full questionnaire in the survey

- The first paragraph under 5.2 actually is "Procedure" and should be distinct from "Measures"

- What is the reference outlining the theoretical basis for the five dimensions?

- what is the argument for doing the CFA in the light of prior empirical tests?

- Was there a specific process for item development?

- The item for trust assessment at the micro level needs to be discussed, scientists "in general" do not reflect individuals

- What is the internal consistency of willingsness to be vulnerable (table2)

- What did coders do if they did not overlap?

- Where are the summaries of the qualitative data (open responses)?

- Could the results by affected by bad compliance (straightclickers) of the sample. What was done for the quality of the data collection?

Results

- please do not write about "significant differences", either there are differences (because significant) or there could not be confirmed differences

- Table 3 . 4.62 (.06) -SD could be a typo here; bold indicates the "highest values"

- l.457: How many used the option?

Discussion

- The argument from 534-36 to 537-39 is not clear

- l.585-587 is too speculative

- l.677/678 how many had high level of attentiveness

- Please discuss: How might the groups' differential propensity for/against science may have affected responding and, so, your results.

- Please discuss: Was the latent profile analysis successful, if you ended up with a distinction based on the actual categories of one item?

- Why do you consider high perception of negative shifts to be an artifact, why not the same about stable trust?

Figure 1

- Uncertainty needs to be indicated with error bars

- Capitalisation needs to be consistent

- the y-axis does not appear to be depicted

Figure 2

- Uncertainty needs to be indicated with error bars

- Capitalisation needs to be consistent

Figure 3

- Capitalisation needs to be consistent

Reviewers' comments:

Reviewer's Responses to Questions

**Comments to the Author**

Reviewer #1: All comments have been addressed

Reviewer #2: All comments have been addressed

Reviewer #3: All comments have been addressed

2. Is the manuscript technically sound, and do the data support the conclusions?

Reviewer #1: Yes

Reviewer #2: Yes

Reviewer #3: Yes

3. Has the statistical analysis been performed appropriately and rigorously?

Reviewer #1: Yes

Reviewer #2: Yes

Reviewer #3: Yes

4. Have the authors made all data underlying the findings in their manuscript fully available?

Reviewer #1: Yes

Reviewer #2: Yes

Reviewer #3: Yes

5. Is the manuscript presented in an intelligible fashion and written in standard English?

Reviewer #1: Yes

Reviewer #2: Yes

Reviewer #3: Yes

Reviewer #1: Dear Authors,

thank you for resubmitting the revised manuscript. The manuscript “Trust in science during the COVID-19 pandemic: A typology of internet users in South Africa” focuses on identifying subgroups of South African internet users with varying degrees of trust in science. Using LPA and reasons to trust science (e.g. expertise, transparency) as the basis for classification, four subgroups were identified. These subgroups were subsequently described in detail and examined regarding different forms of contact, personal experiences with science during COVID-19 pandemic, and perceived changes in trust in science.

The manuscript has benefited considerably from the revisions and now demonstrates a clear and coherent structure. All previously raised issues have been addressed satisfactorily and, where necessary, discussed in appropriate detail. I recommend the manuscript for publication, subject to the following minor adjustments:

Content

- Optional: It may be worth considering whether the limitations could be integrated into the discussion/conclusion section, thereby allowing the manuscript to end with a concise summary or concluding remark.

Statistics

- ANOVAs: Please provide more statistical details (i.e., F statistics, degrees of freedom, p values)

- Please use a capital “N” when referring to the total sample size (see, for example: Abstract l.9; full text ll. 239, 245, 279, 318, 360, 414, 420, 434, 470).

Miscellaneous

- L. 228 – clarification required: 18 “days”?

- l. 233 – “Ethics”

- l. 235/236 – it might be advisable to place this statement and the corresponding reference in a footnote.

- l. 645 – the first sentence should be completed.

Reviewer #2: The authors have responded thoroughly to my previous comments. The introduction is now clearer, with improved integration of relevant literature and explicit research objectives. The methods and results sections were strengthened by clarifying the latent profile analysis procedures, adding effect sizes, and providing richer descriptions of group differences and contact types. The discussion was reorganized to improve coherence, better connect findings to theory, and include cultural specifics and policy implications. Open-ended responses were analyzed more systematically, and limitations were more transparently discussed.

Overall, the revisions have addressed my concerns and substantially improved the manuscript. I have no further major comments.

Reviewer #3: Thank you for the opportunity to review this revised manuscript. The additions to the Introduction and Discussion, and the reorganization of parts of these sections has much improved the manuscript and the intent of the research. I commend the authors on the extensive revisions to these sections.

The changes to the description of each typology is much improved. Likewise, the more nuanced discussion of the limitations of the research is a welcome addition.

Thank you for clarifying the scale is based on another theoretical scale, hence the use of CFA. However, it is worth noting that this scale has been validated in a WEIRD sample, and may not necessarily translate to a non-WEIRD sample. It would be beneficial to use EFA to identity data-driven factors, and then consider the use of CFA to compare this data-driven model against the five-factor model tested. While validation of the scale is not the focus of the paper it would be useful to see this in the Supplementary Materials – or a direction for future research (i.e., cross-cultural validation of this scale). Having said that not doing so would not preclude me from recommending accept on the current version. This is just a suggestion.

Typo line 135, tent should probably be “tend”.

**Do you want your identity to be public for this peer review?** For information about this choice, including consent withdrawal, please see our Privacy Policy

Reviewer #1: **Yes:**  Holger Futterleib

Reviewer #2: No

Reviewer #3: No

---

## [Author Response · Author response to Decision Letter 2]

10 Dec 2025

please see document for responses

---

## [Editor Report · Decision Letter 2]

29 Dec 2025

Trust in science during the COVID-19 pandemic: A typology of internet users in South Africa

PONE-D-24-17791R2

Dear Dr. Reif,

We’re pleased to inform you that your manuscript has been judged scientifically suitable for publication and will be formally accepted for publication once it meets all outstanding technical requirements (see my comments below).

Kind regards,

Felix G. Rebitschek

Academic Editor

PLOS One

Additional Editor Comments (optional):

Dear Authors,

Thanks for concisely addressing the comments!

In the light of the acceptance, my last requirements are as follows:

Regarding your question: Reference to the summary of open-ended responses with the help of the link to the repository is fine.

Regarding the structure:

2 and 3 should be part of 1 Introduction

The structuring of the subsections under Methods is still not resolved. Let me suggest an alternative approach:

2.1 Sample: currently this is in l. 238 - 245 and l. 259 - 276

2.2 Material and measures: currently in l. 246 - 249, 278 - 350

2.3 Procedure: this would be from currrent l. 249 - 258

2.4 Analysis: currently l. 351+
---

## [Editor Report · Acceptance letter]

PONE-D-24-17791R2

PLOS One

Dear Dr. Reif,

I'm pleased to inform you that your manuscript has been deemed suitable for publication in PLOS One. Congratulations! Your manuscript is now being handed over to our production team.

Kind regards,

on behalf of

Dr. Felix G. Rebitschek

Academic Editor

PLOS One